# Comparison of Physical and Biochemical Characterizations of SARS-CoV-2 Inactivated by Different Treatments

**DOI:** 10.3390/v14091938

**Published:** 2022-08-31

**Authors:** Shouzhi Yu, Yangyang Wei, Hongyang Liang, Wenheng Ji, Zhen Chang, Siman Xie, Yichuan Wang, Wanli Li, Yingwei Liu, Hao Wu, Jie Li, Hui Wang, Xiaoming Yang

**Affiliations:** 1Beijing Institute of Biological Products Company Limited, Beijing 100176, China; yushouzhi@sinopharm.com (S.Y.); yangyangwei3391@163.com (Y.W.); ayangyang@163.com (H.L.); 18298466258@163.com (W.J.); 13401114174@163.com (Z.C.); xsiman2017@163.com (S.X.); w250658055@163.com (Y.W.); lolired0704@163.com (W.L.); lywhing@126.com (Y.L.); wuhaobio@163.com (H.W.); lijie10201996@163.com (J.L.); 2China National Biotec Group Company Limited, Beijing 100024, China

**Keywords:** SARS-CoV-2, virus inactivation, formaldehyde, β-propiolactone, S protein, surface plasmon resonance

## Abstract

Severe acute respiratory syndrome coronavirus 2 (SARS-CoV-2) has caused huge social and economic distress. Given its rapid spread and the lack of specific treatment options, SARS-CoV-2 needs to be inactivated according to strict biosafety measures during laboratory diagnostics and vaccine development. The inactivation method for SARS-CoV-2 affects research related to the natural virus and its immune activity as an antigen in vaccines. In this study, we used size exclusion chromatography, western blotting, ELISA, an electron microscope, dynamic light scattering, circular dichroism, and surface plasmon resonance to evaluate the effects of four different chemical inactivation methods on the physical and biochemical characterization of SARS-CoV-2. Formaldehyde and β-propiolactone (BPL) treatment can completely inactivate the virus and have no significant effects on the morphology of the virus. None of the four tested inactivation methods affected the secondary structure of the virus, including the α-helix, antiparallel β-sheet, parallel β-sheet, β-turn, and random coil. However, formaldehyde and long-term BPL treatment (48 h) resulted in decreased viral S protein content and increased viral particle aggregation, respectively. The BPL treatment for 24 h can completely inactivate SARS-CoV-2 with the maximum retention of the morphology, physical properties, and the biochemical properties of the potential antigens of the virus. In summary, we have established a characterization system for the comprehensive evaluation of virus inactivation technology, which has important guiding significance for the development of vaccines against SARS-CoV-2 variants and research on natural SARS-CoV-2.

## 1. Introduction

The rapid spread of the coronavirus disease 2019 (COVID-19), which is caused by severe acute respiratory syndrome coronavirus 2 (SARS-CoV-2), has taken more than 6 million human lives and caused huge economic losses around the world [1]. The SARS-CoV-2 infection has a wide range of clinical manifestations, ranging from asymptomatic to dysregulated immune response and severe/deadly disease [2,3,4]. Due to its fast-spreading nature and a lack of effective therapeutics, live SARS-CoV-2 virus research must be conducted in a biosafety level 3 (BSL3) facility [5]. In laboratories with lower biosafety levels and in the development of all types of vaccines derived from whole viral particles, the SARS-CoV-2 virus must be completely inactivated and rendered non-infectious. The complete inactivation of SARS-CoV-2 can help to ensure that experimental personnel work under safe conditions, increase the number of laboratories and researchers engaged in SARS-CoV-2-related research, and ensure the safety of inactivated virus-related biological products such as inactivated vaccines [6]. Vaccines prepared via the inactivation of the whole virus play an important role in the global fight against COVID-19, as they account for half the global vaccine supply to date [7,8].

An appropriate inactivator, correct inactivation procedures, and inactivation validation are the most critical factors in the production of inactivated vaccines [9,10,11,12]. A reasonable inactivation procedure should be tailored to the virus strain, the inactivator concentration, and many other factors such as incubation time and virus concentration must be considered to achieve a functional product [12]. Viral inactivation can be achieved by physical (heat, UV irradiation) and chemical methods (detergents, SDS, TRIzol, formaldehyde, and BPL) [6,13]. Formaldehyde and BPL are the most readily available chemical inactivators for the inactivation of viruses, and both are approved by regulatory agencies in various countries to produce virus-inactivated biological products [14]. Formaldehyde inactivates viruses through the cross-linking of virus surface proteins, while BPL inactivates viruses mainly via the acylation or alkylation of virus DNA or RNA [15]. In the process of preparing COVID-19 vaccines, BPL and formaldehyde are used to inactivate SARS-CoV-2 virus particles [7,16,17,18]. However, because this is a new virus, there is currently no comprehensive study on the effects of formaldehyde and BPL inactivation processes on the physical and chemical properties of SARS-CoV-2, and there is no scientific standard for quantitatively evaluating the effects of inactivation processes.

Surface plasmon resonance (SPR), a technology that allows for the label-free and real-time detection of antigen–antibody interactions, has been used to detect antibodies that are specific against SARS-CoV-2 [19]. Circular dichroism (CD) spectroscopy detects changes in proteins’ secondary structures using UV radiation and has been used to detect the S1 spike protein receptor-binding domain of SARS-CoV-2 [20]. In this study, we comprehensively assessed the effects of four different chemical inactivation methods (formaldehyde, formaldehyde + BPL, BPL, BPL + BPL) on the morphology, physical properties, and biochemical properties of SARS-CoV-2 using size exclusion chromatography, western blotting, ELISA, an electron microscope, dynamic light scattering, circular dichroism, and surface plasmon resonance. We established an effective evaluation system for the chemical inactivation of SARS-CoV-2 and confirmed that the BLP inactivation is a better process than the other methods in this study. We observed that the inactivation method of BPL treatment for 24 h not only ensures the complete inactivation of the virus but also maintains the maximum homogeneity of the virus particles and the integrity of the antigen. Given that the quality of the inactivated virus largely depends on the choice of inactivator, this evaluation of the SARS-CoV-2 inactivation process can guide our preparation of inactivated virus samples and the optimization of the process during vaccine production.

## 2. Materials and Methods

### 2.1. Virus Culturing

The virus strain was obtained from the Beijing Institute of Biological Products Co., Ltd., (batch number: W2021 (COVID-19-BJ-P7)03). The viruses were cultured in a 10 L basket bioreactor at a temperature of 36 ± 1 °C. The virus supernatant was harvested 48–72 h after inoculation.

### 2.2. Virus Inactivation

After harvesting, the virus supernatant was clarified using a 1.5/0.8 μm filter. The clarified virus supernatant was then inactivated using one of the four following methods. (a) The virus was treated with formaldehyde at 36.5 ± 1 °C and a concentration of 0.1 g/L for 48 h then dialyzed with 0.02 M PBS for 4 h three times to remove the formaldehyde. (b) The virus was treated with formaldehyde at 36.5 ± 1 °C and a concentration of 0.1 g/L for 48 h, dialyzed with 0.02 M PBS for 4 h three times, and then treated with BPL at a concentration of 1:4000 at 2–8 °C for 20–24 h. The BPL was then degraded at 37 ± 1°C for 2 h. (c) The virus was inactivated by BPL at a concentration of 1:4000 for 20–24 h at 2–8 °C. The BPL was then degraded at 37 ± 1 °C for 2 h. (d) The virus was inactivated with BPL at a ratio of 1:4000 at 2–8 °C for 20–24 h, followed by cell debris clarification, incubation at 37 ± 1 °C for 2 h, and a second BPL inactivation (1:4000 (*v*/*v*)).

### 2.3. Validation of the Inactivation

Ten milliliters of inactivated SARS-CoV-2 was used to inoculate Vero monolayers in 75 cm^2^ flasks, and negative control cells were prepared; the cells were then cultured at 36 ± 1 °C for 4 days. This was the first passage. Then, 20 mL of supernatant from the cells in the flask was inoculated onto two additional Vero monolayers in 75 cm^2^ flasks (10 mL each) and incubated at 36 ± 1 °C for 4 days. This was the second passage. Then, 10 mL of supernatant from the cells in the flask was inoculated onto further Vero monolayers in 75 cm^2^ flasks and incubated at 36 ± 1 °C for 4 days. This was the third passage. No cytopathic effect (CPE) was observed in any of the three passages.

After being consecutively blind passaged for three generations, the supernatant and cells were re-verified by immunofluorescence. The supernatant was inoculated onto Vero cells and cultured in a 37 °C incubator. The cells were cultured in a 12-well plate after digestion with trypsin for 48 h and then washed three times with potassium-free PBS. The cells were fixed with 4% paraformaldehyde, and primary antibody (rabbit anti-SARS-CoV-2, dilution 1:100) was added overnight at 4 °C, then they were washed three times with PBS. The secondary antibody (FITC-labeled anti-rabbit IgG, dilution 1:100) was added for 40 min at 37 °C then washed three times with PBS. Then, DAPI dye for nuclear staining was added to each sample for 5 min, after which they were washed with PBS. The cells were mounted with PBS–glycerol (1:1) solution and examined with a fluorescence microscope.

### 2.4. Virus Purification

The inactivated SARS-CoV-2 viral supernatant was filtered through a 0.45 μm filter membrane to remove cell debris. Then, concentrated virus was obtained via 30 times ultrafiltration concentration using filters with 300 kDa cut-off membranes. Nuclease was added to a final concentration of 50 U/mL at 37 °C for 1 h to degrade the residual DNA in the Vero cells. Capto Core 700 gel was loaded into the column, washed with 1 mol/L sodium hydroxide solution, and balanced with PBS buffer. The concentrated virus was loaded into the gel and eluted with PBS. Under the detection of UV 280 nm, the first peak was collected for the follow-up experiment.

### 2.5. SDS-PAGE

The purity chromatography was characterized by 12% protein electrophoresis preformed on M5 Prestained Plus Protein Ladders. The images were analyzed using a luminescent image analyzer.

### 2.6. Western Blot

The protein samples were denatured via boiling for 10 min and loaded onto SDS-PAGE for electrophoresis. The gel was transferred to a PVDF membrane at 50 V for 2 h and closed using 5% BSA, followed by the addition of 1/1000-diluted SARS-CoV-2 spike primary antibody (40591-T62, Sino Biological, Beijing, China) overnight. It was then washed three times using PBST (PBS: Tween = 1000:1), incubated for 1 h with 1/10,000-diluted IgG secondary antibody (17294442, Cytiva, Marlborough, MA, USA), washed three times using PBST, and incubated using ECL. The luminescent image was detected using Amersham ImageQuant 800.

### 2.7. Enzyme-Linked Immunosorbent Assay (ELISA)

Coated antibodies (20210310, BIBP, Beijing, China) were diluted using 1/5000 coating solution and placed on ELISA plates at 2–8 °C overnight at 100 μL/well. They were washed five times with 300 μL/well of PBST (PBST:Tween = 2000:1), and 200 μL/well of PBS containing 1% BSA was added for 2 h at 37 °C. The closure solution was then decanted for use. We added 100 μL/well of standard, negative control and inactivated sample and to ELISA plates at 37 °C in a constant temperature water bath for 2 h. The standard inactivated virus is the reference product for the determination of the antigen content of the SARS-CoV-2 inactivated vaccine (Vero cells) stock solution (STD-nCoV-2021004, BIBP, China). The standard, the antigen concentration of which was 16 U/mL, was diluted from 2 to 128 times. The samples were diluted from 2 to 256 times. Then, the plate was washed with PBST five times. We added 100 μL/well of 1/1250-diluted antibody detection working solution (W2021032S, BIBP, China) and incubated the solution for 1 h at 37 °C in a constant temperature water bath. We then washed it five times with PBST. We added 100 μL/well of 1/4000-diluted enzyme labeled antibody working solution and incubated it for 1 h in a constant temperature water bath at 37 °C. We then washed it five times with PBST. We added 50 μL/well of color development A and color development B solution, respectively, and incubated them for 15 min at room temperature while protected from light. We added 50 μL/well of the termination solution and read the reaction at 450/630 nm with an enzyme marker within 15 min of termination. Four-parameter fitting was performed on the standard concentration and absorbance to obtain the standard curve equation using Gen5 software.

### 2.8. Quantitative Determination of Protein

The protein concentration of the virus samples was measured via the Lowry assay. To 1 mL of sample and protein standards from 5 to 100 μg/mL, we added 1 mL of the alkaline copper reagent, and then, we mixed it and allowed it to stand for 10 min. We added 4 mL of Folin–Ciocalteu’s reagent mix diluted 16 times, vortexed it thoroughly, and incubated it for 30 min. After incubation, we vortexed it again and measured the absorbance at 650 nm.

### 2.9. EM Sample Preparation

Samples were applied to a carbon-coated copper grid previously emitted at low pressure, stained with 2% uranyl acetate, and visualized with a Tecnai T12 electron microscope (FEI) equipped with a LaB6 filament operating at an accelerating voltage of 120 kV. The images were recorded on imaging plates using a low-dose procedure with a magnification of 23,000× and a bokeh of approximately 1.5 μm.

### 2.10. Dynamic Light Scattering (DLS)

The average size and zeta potential of BNP, DNP, MNP, and CNP were determined by dynamic light scattering (DLS) using a ZetaSizer Nano Series Nano ZS (Malvern Instruments Ltd., Malvern, UK). Measurements were performed at a constant angle of 173° after appropriate dilution of inactivated samples in distilled water. Each batch was analyzed three times.

### 2.11. Circular Dichroism (CD) Spectroscopy

The protein concentration of the inactivated sample was fixed at 0.5 mg/mL. The circular dichroism (CD) was obtained using a MOS-500 circular dichroism chromatograph with the following parameters: starting wavelength 190 nm; ending wavelength 260 nm; step 1 nm; repeat one time; acquisition period 1s/point; UV spectrum from 190 to 260 nm. The secondary structures of the samples were fitted and calculated using CDNN software, and the results reported include Helix (Helix), Antiparallel (Anti-parallel β-fold), Parallel (Parallel β-fold), Beta-Turn (Beta-turn), and Rndm. Coil (Irregular Coil).

### 2.12. Surface Plasmon Resonance (SPR)

Surface plasmon resonance (SPR) experiments were performed using a BIAcore T200 machine (BIAcore, GE Healthcare) in PBS buffer (with 0.05% Tween added) at 25 °C. The purified virus was immobilized on the surface of the CM5 sensor chip using the NHS/EDC method, resulting in a response unit of approximately 5000. A gradient concentration of SARS-CoV-2 spike chimeric monoclonal antibody (40150-D002, Sino Biological, Beijing, China) was then passed through the chip at a rate of 30 μL/min. After each injection cycle, the chip was regenerated using a solution containing 3 M MgCl_2_. Binding affinity was obtained by globally fitting the curve using BIA evaluation software.

## 3. Results

### 3.1. Effects of Different Inactivation Methods on the Homogeneity of Virus Particles

The infectivity of the viral supernatants was measured by CPE. Our results demonstrate that all four inactivation methods were efficient at inactivating the virus completely (data not shown). Firstly, we purified the virus and examined the homogeneity of the viral particles treated with different inactivation methods using size exclusion chromatography. The results showed that the UV absorption curves of formaldehyde-inactivated and formaldehyde + BPL-inactivated groups were wide and flat (Figure 1A,B). The UV absorption curves of the BPL-inactivated and BPL + BPL-inactivated groups were narrow and sharp (Figure 1C,D). These results indicate that formaldehyde-inactivated and formaldehyde + BPL-inactivated treatments affect the dispersity and homogeneity of the viral particles. The remaining two inactivation methods had less of an impact on the physical properties of the viral proteins.

### 3.2. Formaldehyde and BPL Treatment Have Less of an Impact on the Morphology of SARS-CoV-2

Electron microscopy has been used to evaluate the structural characteristics of inactivated SARS-CoV-2 [21]. To test whether SARS-CoV-2 undergoes morphologic changes during formaldehyde or BPL treatment, we used an electron microscope to study the virus morphology. Viral supernatant treated with formaldehyde or BPL was analyzed via negative staining. As is shown in Figure 2A–D, the virus particles treated with different inactivation methods have normal morphological structures and crown shapes. These results indicate that formaldehyde and BPL treatment have little impact on the morphological structure of SARS-CoV-2.

### 3.3. Long Time Treatment of BPL Causes the Aggregation of SARS-CoV-2

Earlier studies have demonstrated that BPL treatments with high concentrations cause the aggregation of virus particles [9] and that formaldehyde treatment causes protein crosslinking [15]. To test whether SARS-CoV-2 undergoes aggregation during formaldehyde or BPL treatment, we used dynamic light scattering (DLS) to study the virus’s particle size distribution in a suspension (Figure 3A–D). The dynamic light scattering results showed that the viral particles treated with BPL for 24 h had an average size of about 127 nm and that the size increased to over 150 nm after 48 h of treatment with BPL (Table 1). The particle size of the BPL-inactivated group was significantly lower than that of the BPL + BPL group and formaldehyde + BPL group (Figure 3E). The particle size of the BPL + BPL-inactivated group was significantly greater than that of the other three groups (Figure 3E). Taken together, these results demonstrate that treatment with BPL for a long time causes the aggregation of SARS-CoV-2 particles in a time-dependent manner. Therefore, it is necessary to control the inactivation time when using BPL (not higher than 24 h) and to remove BPL quickly after inactivation.

### 3.4. Formaldehyde and BPL Treatment Have Less of an Impact on the Secondary Structure of SARS-CoV-2

We used CD analyses in the far region to estimate the secondary structure of the virus (Figure 4A–D). According to a quantitative analysis of the far UV-CD spectra, as can be seen in Table 2, the viruses treated with the four different inactivation methods have similar proportions of α-helixes, antiparallel β-sheets, parallel β-sheets, β-turns, and random coils, indicating that the treatments only have a tiny impact on the secondary structure of SARS-CoV-2. These results suggest that formaldehyde and BPL treatment have no significant effect on SARS-CoV-2’s secondary structure.

### 3.5. Effects of Different Inactivation Methods on the Content of S Antigen

We further tested the epitope integrity of SARS-CoV-2 virions inactivated with BPL. Virus samples were electrophoresed via SDS-PAGE to visualize the antigens. SDS-PAGE and western blotting results indicated that the formaldehyde-inactivated and formaldehyde + BPL-inactivated treatments caused crosslinks and viral protein aggregation, as can be seen in Figure 5A,B. However, BPL inactivation does not have this adverse effect (Figure 5A). We then studied the effect of different inactivation methods on virus antigens via ELISA. The results showed that BPL-inactivated virus particles had the highest antigen/protein ratio when compared with the other groups (Table 3). Formaldehyde treatment caused a considerable loss of antigens. In summary, the above results indicate that BPL inactivation can better maintain virus antigenic integrity.

### 3.6. Formaldehyde and BPL Treatment Does Not Affect the Ability of the SARS-CoV-2 to Bind to the S Protein Antibody

The SPR method is commonly used for detecting antigen–antibody affinity [22]. To further evaluate the impact of different inactivation methods on the immunological properties of SARS-CoV-2, we performed SRP experiments. We used the SARS-CoV-2 spike chimeric monoclonal antibody to examine the equilibrium dissociation constants for viruses treated with four different inactivation methods. The results showed that the affinities of the viruses treated by the four inactivation methods to the SARS-CoV-2 spike chimeric monoclonal antibody D002 were all at the nanomolar level, with no significant differences between them (Figure 6A–D, Appendix A). These results indicate that formaldehyde and BPL treatment made no significant difference to the affinity of SARS-CoV-2 for S monoclonal antibody.

## 4. Discussion

Due to the huge loss of life and property caused by COVID 19, the development of effective vaccines is a top priority. Several vaccine platforms are being investigated, including whole virus vaccine platforms, protein-based vaccine platforms, and nanoparticle and genetic vaccine platforms [1].

Virus inactivation is one of the most important steps in virology research and inactivated vaccine development. Common inactivators include β-propiolactone and formaldehyde. However, there is no common standard for the design and determination of the inactivation process. At present, intensive methods are usually selected in laboratory studies, such as heat, detergent, UV, high-concentration formaldehyde, etc., which may cause irreversible damage to the virus [6,23]. In vaccine production, the inactivator BPL, which was applied in the first SARS-CoV-2 inactivated vaccine, has been used [7,24,25,26]. Meanwhile, there have also been reports on the combination of formaldehyde and BPL to inactivate the virus. However, the impact on various aspects of the nature of the virus was not evaluated in a systematic way during the selection of inactivators.

Because BPL and formaldehyde are both common inactivators and have been reported to inactivate SARS-CoV-2 successfully, four inactivation methods were reproduced in this study, all of which inactivated the SARS-CoV-2 virus thoroughly. Then, we used various tools to characterize the physical and biochemical properties of SARS-CoV-2 viral particles following different treatments.

The results showed that formaldehyde treatment reduced the homogeneity of virus particles. Although there were no significant differences in the negative staining or circular dichroism spectra, the treament’s performance in terms of size exclusion chromatography became worse, and its UV absorption curve became less sharp. An examination of the S protein showed that the viral S protein after formaldehyde treatment displayed dispersive bands and cross-linking upon SDS-PAGE. Compared with the 20–24 h BPL treatment, the S antigen content decreased by more than twice. After BPL treatment, the virus particles remained relatively homogenous, and the S protein was still clear. These results confirm the chemical properties of formaldehyde, which can cause viral protein cross-linking and denaturation and damage viral surface antigens. Therefore, although formaldehyde does not destroy the morphology or secondary structure of the virus, compared with BPL, formaldehyde is not a suitable SARS-CoV-2 inactivator.

It is noteworthy that BPL treatment over a long time (48 h) can cause aggregation so that the virus particle sizes increased from about 130 nm to 150 nm, and the surface S antigen content is also slightly reduced. Previous studies have shown that high concentrations of BPL also cause the aggregation of SARS-CoV-2 [9]. Therefore, attention should be paid to treatment concentration and treatment time during BPL inactivation.

This study also attempted to study the function of the S protein in inactivated SARS-CoV-2. We tested the equilibrium dissociation constants of viral particles treated with different inactivators with monoclonal antibodies against the S protein, and the results showed only tiny differences in their affinity for antibodies. This may be due to the limitations of SPR. Formaldehyde treatment can destroy the S protein and lead to a decrease in the antigen amount, but it cannot completely destroy the antigen count. The remaining S protein still maintained a good and consistent affinity for monoclonal antibodies.

Previous studies have shown that inactivators affect virus antigenicity, particle size, and surface protein functions (such as the affinity for receptors, the ability to promote membrane fusion, etc.) [9,14,15]. Our study confirmed that different inactivators and inactivation procedures also have effects on SARS-CoV-2 in terms of its physical and biochemical characteristics.

In this study, the effect of different inactivation processes on SARS-CoV-2 was comprehensively assessed. We found that 20–24 h BPL treatment can best ensure the complete inactivation of the virus, retain antigen activity to the maximum extent, and ensure virus dispersity. Additionally, we used a variety of systematic and effective tools to characterize the inactivated viruses in a multi-dimensional manner. These results provide important guidance and evaluation tools for SARS-CoV-2 research and for the selection and continuous optimization of inactivators and process parameters in vaccine production.

## Figures and Tables

**Figure 1 viruses-14-01938-f001:**
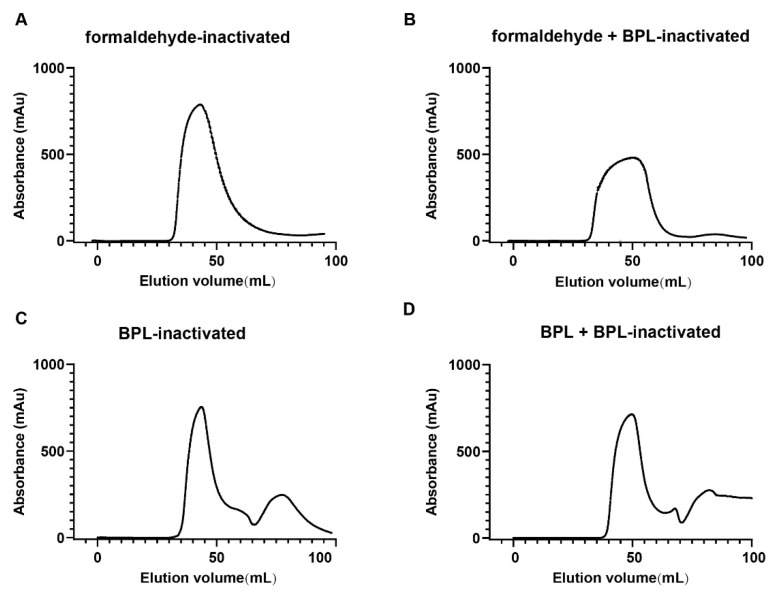
Size exclusion chromatography results of different inactivated virus samples (**A**–**D**). Size exclusion chromatography (SEC) results of viral supernatant inactivated with four different chemical inactivation methods (**A**) Formaldehyde-inactivated. (**B**) Formaldehyde + BPL-inactivated. (**C**) BPL-inactivated. (**D**) BPL + BPL-inactivated.

**Figure 2 viruses-14-01938-f002:**
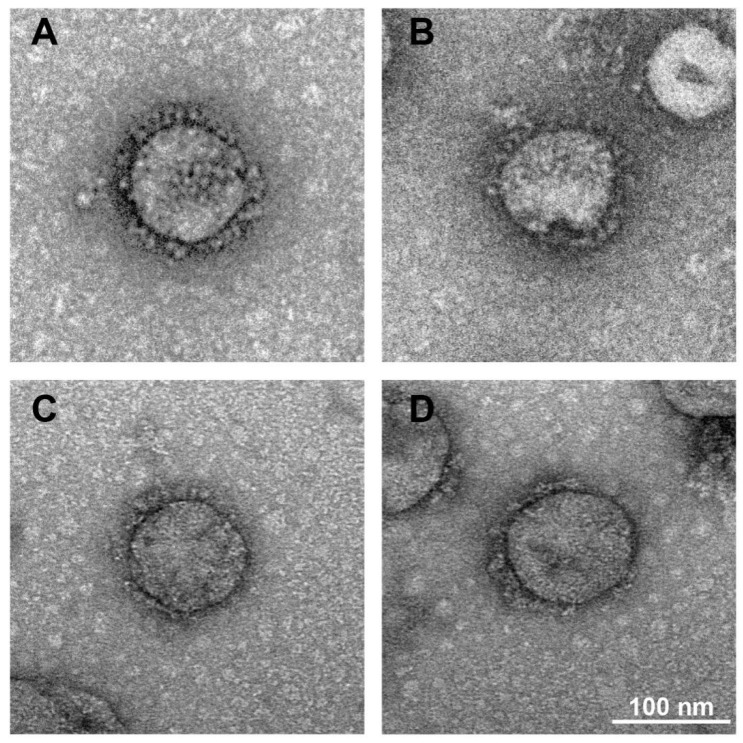
Electron microscope images of the morphology of different inactivated virus samples. (**A**) Formaldehyde-inactivated. (**B**) Formaldehyde + BPL-inactivated. (**C**) BPL-inactivated. (**D**) BPL + BPL-inactivated. Scale bar, 100 nm.

**Figure 3 viruses-14-01938-f003:**
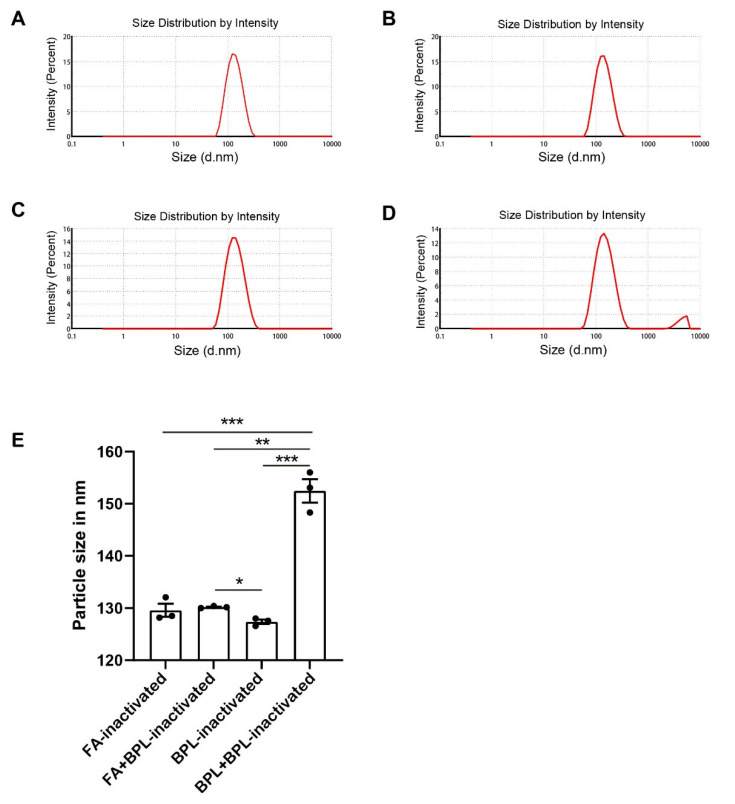
Dynamic light scattering (DLS) analysis graphs of different inactivated virus samples: (**A**) Formaldehyde-inactivated (FA-inactivated). (**B**) Formaldehyde + BPL-inactivated (FA + BPL-inactivated). (**C**) BPL-inactivated. (**D**) BPL + BPL-inactivated. (**E**) Dynamic light scattering (DLS) results of different inactivated virus samples. FA, formaldehyde. * *p* < 0.05, ** *p* < 0.01 and *** *p* < 0.001 were defined as statistically significant.

**Figure 4 viruses-14-01938-f004:**
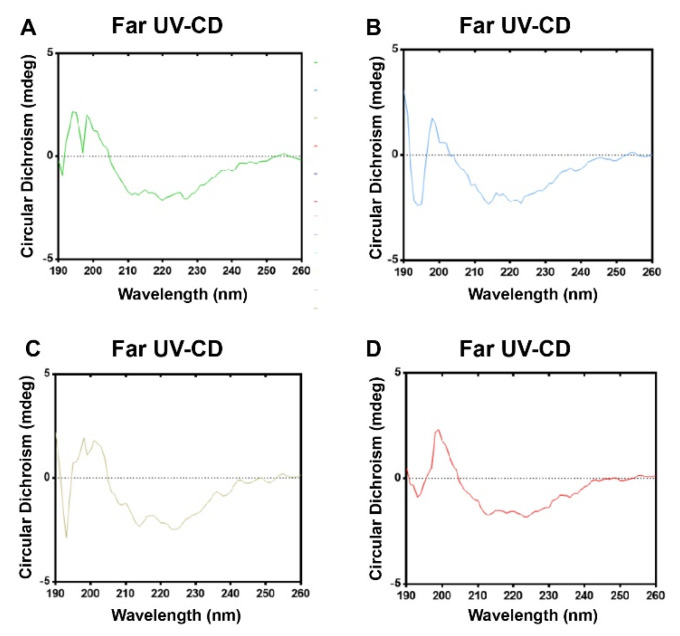
Circular dichroism (CD) spectroscopy analysis charts of different inactivated virus samples. (**A**) Formaldehyde-inactivated virus samples. (**B**) Formaldehyde + BPL-inactivated virus samples. (**C**) BPL-inactivated virus samples. (**D**) BPL + BPL-inactivated virus samples.

**Figure 5 viruses-14-01938-f005:**
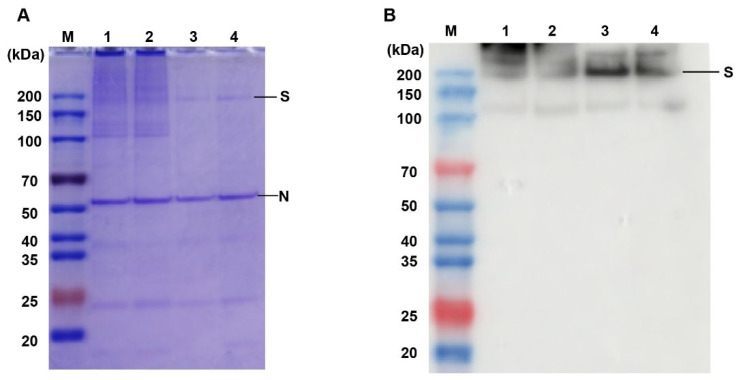
SDS-PAGE and western blot analyses of inactivated virus samples. (**A**) SDS-PAGE of inactivated virus samples, (**B**) Western blot analyses of inactivated virus samples. M: protein ladder; 1: formaldehyde-inactivated; 2: formaldehyde + BPL-inactivated; 3: BPL-inactivated; 4: BPL + BPL-inactivated.

**Figure 6 viruses-14-01938-f006:**
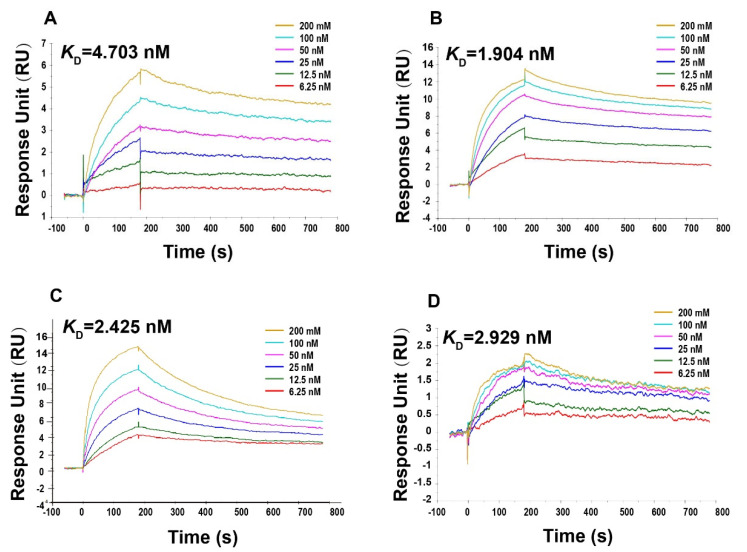
Surface plasmon resonance (SPR) analysis graphs of different inactivated virus samples to SARS-CoV-2 Spike chimeric monoclonal antibody D002. (**A**) Formaldehyde-inactivated. (**B**) Formaldehyde + BPL-inactivated. (**C**) BPL-inactivated. (**D**) BPL + BPL-inactivated.

**Table 1 viruses-14-01938-t001:** Dynamic light scattering (DLS) results of different inactivated virus samples.

Sample	Particle Size (nm)	PDI (Polydispersity Index)
Formaldehyde-inactivated	129.60 ± 1.77	0.11 ± 0.01
Formaldehyde + BPL-inactivated	130.20 ± 0.16	0.13 ± 0.02
BPL-inactivated	127.40 ± 0.59	0.13 ± 0.03
BPL + BPL-inactivated	152.47 ± 3.18	0.25 ± 0.01

**Table 2 viruses-14-01938-t002:** The percentage of secondary structure contents of different inactivated virus samples obtained in the CD studies.

Samples	α-Helixes	Antiparallel β-Sheets	Parallel β-Sheets	β-Turns	Random Coils
Formaldehyde-inactivated	6.80%	47.30%	3.80%	16.70%	29.40%
Formaldehyde + BPL-inactivated	6.50%	48.20%	3.80%	16.50%	29.60%
BPL-inactivated	6.60%	47.50%	3.80%	16.50%	29.80%
BPL + BPL-inactivated	6.50%	47.60%	3.80%	16.50%	29.70%

**Table 3 viruses-14-01938-t003:** ELISA analysis of protein and antigen concentrations of different inactivated virus samples.

Sample	Protein Concentration (μg/mL)	Antigen Concentration (U/mL)	Ratio (U/μg)
Formaldehyde-inactivated	332.4	267.9	0.81
Formaldehyde + BPL-inactivated	211.33	224.57	1.06
BPL-inactivated	327.56	623.38	1.90
BPL + BPL-inactivated	308.19	372.56	1.21

## Data Availability

Not applicable.

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
