# Peer review of "Comparison of Physical and Biochemical Characterizations of SARS-CoV-2 Inactivated by Different Treatments"

_viruses, 2022, doi:10.3390/v14091938_

Round 1
Author Response
Dear reviewer:
Thank you for your work and valuable comments. We are glad to comprehensively revise our manuscript according to your reports.
The following is our reply to your suggest:
- The English language needs to be extensively revised since in this form some paragraphs are unclear and misunderstandable. In addition, there are too many identical words within the same sentence. Please check the whole manuscript and rephrase with the help of a native speake
We have used the rapid language editing services listed at https://www.mdpi.com/authors/english. The English language has been extensively revised.
- Introduction and Discussion sections, as well as references, could be improved with some epidemiological and clinical data: Please read and valuate these articles: 10.3390/idr14030040; 10.1111/imm.13564; 10.1186/s40794-022-00176-4; 10.3892/wasj.2020.64.
We have thoroughly read the above four articles and cited them in the introduction and discussion sections.
- Lines 41-43: The meaning is understandable but English form is unclear, rephrase please
Line 51: Inactivators
Line 51: Inactivators to inactivate… please change
Line 61: specific antibody against SARS-CoV2
Lines 69-70: Please rephrase the sentence, there are too much identical words (method, inactivation, process)
Line 84: Supernatant, check please
Lines 133-134: please rephrase, it is unclear
Line 215: “high concentration of BPL treatment”, please rephrase
Line 259: Antigens
Lines 287-291: Please rephrase the whole sentence since it is unclear and readers could misunderstand
Line 299: "its performance” clarify the subject, please
Line 301: “showed…showed” please use a synonym
Line 304: “These results…” the sentence is incomplete and unclear, please rephrase and check the meaning
Line 326: Please change “novel coronavirus” with “SARS-CoV2”
Line 330-332: Please rephrase, the sentence is unclear
We are very sorry that these sentence and words are unclear and misunderstandable. We have revised them in the latest version.
I'm not sure if our response is the one you're looking for, but if you have any questions, please don't hesitate to contact us.
Best regards
Hui Wang
Reviewer 2 Report
The manuscript studies the effect of four different viral inactivation methods on SARS-CoV-2 viral particles and protein integrity. The authors used various techniques like DLS, SPR, western blot, NsEM, and CD to perform biophysical and biochemical characterization of virus post inactivation.
Major comments:
1. It will be better if the authors show the CPE data in the manuscript's results section to develop a better understanding for the reader.
2. The electron microscopy results are not presented well. Due to the background noise, the results are not clear for each treatment. The visible quantity of spikes is different for each virus particle. In Figure 2D, the spikes do not look like a regular crown, as shown in Figure 2A. Please justify the results.
3. Write the calculation details of values shown in the table3 in the material and methods ELISA section.
Minor comments:
1. In figure 4, on the Y axis, please correct mdge to mdeg
2. Provide the SPR data with fitted curves along with measured curves
Author Response
Dear reviewer:
Thank you for your work and valuable comments. We are glad to comprehensively revise our manuscript according to your reports.
The following is our reply to your suggest:
- It will be better if the authors show the CPE data in the manuscript's results section to develop a better understanding for the reader
Validation of the inactivation (CPE and immunofluorescence) confirmed that four inactivation methods inactivated SARS-CoV-2 effectively. We cultured and inactivated virus, and validated the inactivation in Biosafety Level 3 (BSL3) facility. It’s a pity that the image of cells after inactivation verification was not collected in this special biological protection state. Because no photo instrument was equipped, we only make relevant experimental records. CPE was not observed in the indicator cells tested by neither virus after inactivation nor negative control.
Ps:
Cytopathic effect or cytopathogenic effect (abbreviated CPE) refers to structural changes in host cells that are caused by viral invasion. CPEs are important aspects of a viral infection in diagnostics and validation of the virus inactivation
- The electron microscopy results are not presented well. Due to the background noise, the results are not clear for each treatment. The visible quantity of spikes is different for each virus particle. In Figure 2D, the spikes do not look like a regular crown, as shown in Figure 2A. Please justify the results
We think that the different quantity of spikes in each particle and the different shape of spikes in 2D was caused by negative strain. Because formaldehyde has the function of cross-linking proteins, S protein had been fixed before negative staining. Therefore, the virus particles maintained the good state and spikes was uniform in shape and abundant under the electron microscope observation. However, after BPL inactivation, spikes were still in a variable state. When the virus suffering a severe environment during negative stain, the conformation of S protein would change, and even be induced to the post-fusion conformation, and some of them fell off, so the quantity of S proteins after BPL inactivation was irregular, and some do not show the classic crown conformation.
Considering the different chemical property of formaldehyde and BPL, we believed that the electron microscopic may not show the original state of virus after inactivation in a way, so we also conducted CD to study the conformational differences of the virus. This also suggests that it is very important to select the appropriate method of virus inactivation for different research purposes.
- Write the calculation details of values shown in the table3 in the material and methods ELISA section
We have supplemented the calculation details of values in the material and methods ELISA section in the latest version.
- In figure 4, on the Y axis, please correct mdge to mdeg
We have corrected the Y axis. We are very sorry for this simple mistake.
- Provide the SPR data with fitted curves along with measured curves
We supplement the SPR data with fitted curves along with measured curves as a supplementary data. Please see the attachment.
I'm not sure if our response is the one you're looking for, but if you have any questions, please don't hesitate to contact us.
Best regards
Hui Wang
